# Analytic optimization of *Plasmodium falciparum* marker gene haplotype recovery from amplicon deep sequencing of complex mixtures

**Zena Lapp[1], Elizabeth Freedman[2], Kathie Huang[2], Christine F. Markwalter[1], Andrew A. Obala[3], Wendy Prudhomme-O'Meara[1,2], Steve M. Taylor[1,2] \***

**1** Duke Global Health Institute, Duke University, Durham, North Carolina, United States of America, **2** Division of Infectious Diseases, School of Medicine, Duke University, Durham, North Carolina, United States of America, **3** School of Medicine, College of Health Sciences, Moi University, Eldoret, Kenya

\* steve.taylor@duke.edu

## Abstract

Molecular epidemiologic studies of malaria parasites and other pathogens commonly employ amplicon deep sequencing (AmpSeq) of marker genes derived from dried blood spots (DBS) to answer public health questions related to topics such as transmission and drug resistance. As these methods are increasingly employed to inform direct public health action, it is important to rigorously evaluate the risk of false positive and false negative haplotypes derived from clinically-relevant sample types. We performed a control experiment evaluating haplotype recovery from AmpSeq of 5 marker genes (*ama1*, *csp*, *msp7*, *sera2*, and *trap*) from DBS containing mixtures of DNA from 1 to 10 known *P. falciparum* reference strains across 3 parasite densities in triplicate (n = 270 samples). While false positive haplotypes were present across all parasite densities and mixtures, we optimized censoring criteria to remove 83% (148/179) of false positives while removing only 8% (67/859) of true positives. Post-censoring, the median pairwise Jaccard distance between replicates was 0.83. We failed to recover 35% (477/1365) of haplotypes expected to be present in the sample. Haplotypes were more likely to be missed in low-density samples with <1.5 genomes/µL (OR: 3.88, CI: 1.82–8.27, vs. high-density samples with $\geq$75 genomes/µL) and in samples with lower read depth (OR per 10,000 reads: 0.61, CI: 0.54–0.69). Furthermore, minority haplotypes within a sample were more likely to be missed than dominant haplotypes (OR per 0.01 increase in proportion: 0.96, CI: 0.96–0.97). Finally, in clinical samples the percent concordance across markers for multiplicity of infection ranged from 40%-80%. Taken together, our observations indicate that, with sufficient read depth, the majority of haplotypes can be successfully recovered from DBS while limiting the false positive rate.

**Data Availability Statement:** Code and data to recreate the analyses and figures in this manuscript can be found at https://github.com/

duke-malaria-collaboratory/haplotype_recovery_experiment. Raw sequences have been deposited under BioProject PRJNA1008913.

**Funding:** This work was supported by the National Institute of Allergy and Infectious Diseases (R01AI146849 to W.P-O. and S.M.T. and K01AI175527 to C.F.M.). The funders had no role in study design, data collection and analysis, decision to publish, or preparation of the manuscript.

**Competing interests:** The authors have declared that no competing interests exist.

## Introduction

Malaria parasite surveillance and molecular epidemiologic studies increasingly employ as a genotyping approach amplicon deep sequencing (AmpSeq) of short polymorphic fragments of parasite DNA to identify haplotypes present in a sample. Depending on which segments of the genome are sequenced, this approach returns haplotypes that can be used to estimate complexity of infection [1], investigate transmission between hosts [2, 3], evaluate the prevalence and incidence of markers of drug resistance [4–6], and classify recurrent infections following drug treatment as reinfections or recrudescences [7, 8]. Similarly, these methods are used in molecular epidemiologic studies of viral and bacterial pathogens [9]. As a result of these diverse use cases of AmpSeq, there is a broad need for practical and empirically-derived approaches to maximize haplotype recovery and mitigate the risks of false genotypes.

Prior groups have evaluated the accuracy of haplotype recovery from mixtures containing DNA from known *P. falciparum* strains across a range of available tools and parameters, and reported that strains present in low proportions are likely to be missed [2, 10] and that false positive haplotypes often have lower read depth [11, 12]. In a large analysis of complex mixtures of up to five reference strains, recovery of two markers was compared using four haplotype calling tools [12]. They found that fewer haplotypes are recovered from samples with less *P. falciparum* DNA, that haplotypes with a lower read count were more frequently false positives, and that the four different haplotype calling tools performed similarly. What remains unexplored by these prior reports are investigations of haplotype recovery from samples with three key features of field studies: i) prepared and processed as dried blood spots (DBS), ii) present across a range of densities reflective of infections that are typically observed in field studies, and iii) harboring genomes from a large range of *P. falciparum* strains, which in natural infections can exceed 15 [2].

We evaluated the accuracy of the recovery of diverse *P. falciparum* haplotypes from DBS harboring simple and complex mixtures of parasite genomes. To do so, we prepared mixtures of up to 10 parasite strains at known proportions and across three parasite density categories, and amplified and sequenced each in triplicate with MiSeq across polymorphic segments of five distinct markers (*ama1*, *csp*, *msp7*, *sera2*, and *trap*). With these reads, we employed an existing tool for haplotype inference to investigate the influence of parasite density, genomic complexity, and haplotype censoring criteria on the removal of false positive haplotypes, the sensitivity and precision of haplotype discovery, inter-replicate variability, and the ability to recover expected haplotypes at each locus.

## Materials and methods

### Mock infection design

We selected five targets of interest in the *P. falciparum* genome that have been used in prior AmpSeq studies [2, 7, 13]: *ama1*, *csp*, *msp7*, *sera2*, and *trap*. We amplified by PCR using the reference primers for each (Table 1) from each of ten reference *P. falciparum* strain genomic DNAs (gDNAs), each obtained from BEI Resources and accompanied by Certificates of Analysis: MRA-102G (3D7), MRA-150G (Dd2), MRA-152G (7g8), MRA-155G (HB3), MRA-159G (K1), MRA-176G (V1/S), MRA-1169G (Tanzania), MRA-915G (FUP UGANDA-Palo Alto), MRA-309G (FCB), and MRA-731G (FCR3/Gambia). The products of each individual strain were Sanger sequenced to determine the reference sequence for each strain.

For each reference strain and marker (n = 50), Unipro UGENE v42 [14] was used to map forward and reverse reads from Sanger sequencing to the respective marker gene. The trimming quality threshold and mapping minimum similarity were set to zero. The sequences

**Table 1. Marker-specific reference primers.**

| Marker | Forward primer | Reverse primer |
|---|---|---|
| *ama1* | TCAGGGAAATGTCCAGTATTTG | GGACCATTATTTTCTTGAGCTG |
| *csp* | TTAAGGAACAAGAAGGATAATACCA | AAATGACCCAAACCGAAATG |
| *msp7* | ATGAACAAGAGATATCAACACA | TTAAATTGTTCATGGTATTCCTTA |
| *sera2* | TACTTTCCCTTGCCCTTGTG | CACTACAGATGAATCTGCTACAGGA |
| *trap* | TCCAGCACATGCGAGTAAAG | AAACCCGAAAATAAGCACGA |

were manually trimmed and, where discrepancies in base calls were observed between forward and reverse reads, bases were called manually. Where possible, the Sanger sequences were validated against publicly-available sequences. These sequences were defined as the true reference sequence for each strain, and this the reference strain haplotypes (n = 5 per strain, 1 for each marker).

Five mock polygenomic infections and a 3D7-only mock infection were created by making control mixtures that combined 1 ng/μl gDNA stocks of the distinct parasite reference strains in known percentages ranging from 1% to 100% (Fig 1A). Each control mixture was serially diluted in uninfected whole blood, and dried blood spots (DBSs) were made for each of the 11 dilutions per mixture. DBS were singly punched into individual wells of a deep 96-well plate, and a modified Chelex-100 protocol [3] was used to make gDNA extracts. These were then tested in duplicate with a duplex pfr364/human β-tubulin quantitative PCR (qPCR) assay that estimated parasite densities using a standard curve generated with extracts from control DBS at dilutions of *P. falciparum* 3D7 ranging from 0.1 to 2000 parasites/μL of whole blood [15].

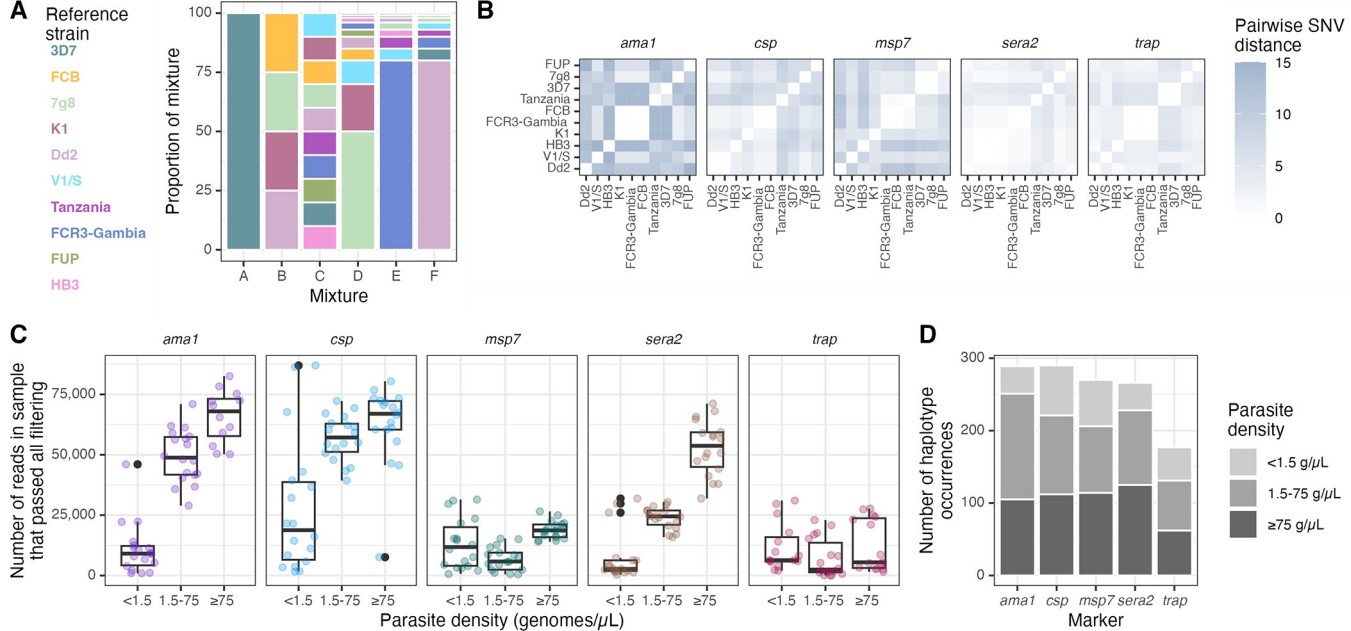

**Fig 1. Overview of mixtures, reference strains, and sequence yield. (A)** Overview of mixtures A through F, each composed of various proportions of gDNA from the listed *P. falciparum* reference strains (colors). **(B)** Pairwise single nucleotide variant (SNV) distances between reference haplotypes of each of the marker genes obtained by Sanger sequencing. **(C)** Number of reads in each sample by parasite density bin, faceted by marker gene. **(D)** Total number of pre-censored haplotype occurrences for each marker across all mixtures and replicates, colored by parasite density bin. Note that *ama1* was sequenced separately from the other markers so read depth cannot be directly compared between *ama1* and other markers. g/μL = genomes/μL.

Control mixture extracts were assigned to one of three parasite density ranges (low, <1.5 genomes/μl; medium, 1.5–75 genomes/μl; and high, ≥75 genomes/μl) and pooled by mixture at each density range for a total 18 pools (6 mixtures x 3 densities) to be used as templates for subsequent PCR amplification.

## Library preparation and sequencing

Each mixture template was prepared for sequencing according to qPCR Ct-value as described in [3]. Then, from each mixture template, we amplified at the target segments of *ama1*, *csp*, *msp7*, *sera2*, and *trap* in individual reactions in triplicate using a nested PCR strategy. Library preparation for sequencing followed described methods [16] with the following exceptions: PCR1 reactions included 300nM of each primer and 7 μl of template gDNA when extract Ct was < 28 (high density), 18 μl when 28 ≤ Ct < 34 (medium density), and 15 μl concentrated extract when Ct ≥ 34 (low density). PCR 1 cycling conditions were 95C x 3′ → (98C × 20s → 62C × 15s → 72C × 20s) × 8 → (98C × 20s → 70C × 15s → 72C × 20s) x 27 → 72C × 1′. PCR 2 reactions included 2 μl of template when gDNA pool extract Ct was < 28, and 8 μl of template when Ct was ≥ 28. The resulting dual-indexed libraries were then pooled and purified as previously described [16] before sequencing on an Illumina MiSeq (v3 300PE) platform. Raw sequences have been deposited under BioProject PRJNA1008913.

## Haplotype recovery

We used Snakemake v 7.20.0 [17] to build an integrated pipeline for haplotype recovery, BRAVA (Basic and Rigorous Amplicon Variant Analyzer; https://github.com/duke-malaria-collaboratory/BRAVA) in order to trim, filter, and map reads, and thence call haplotypes. Primers and adapters of amplicon deep sequencing reads for each marker were removed using Cutadapt v4.1 [18]. These reads were trimmed using Trimmomatic v0.38 [19]; this removed the leading and trailing bases below a Phred quality score of 10, removed all nucleotides from the 3' end after the quality of the read falls below an average Phred quality score of 15 over a sliding window of 4 nucleotides, and dropped reads with fewer than 80 nucleotides. Remaining reads were mapped to the 3D7 reference genome using BBmap v39.01 [20]. In a sensitivity analysis, the mapping was repeated using the HB3 reference genome. Reads were then further filtered and trimmed using the R package DADA2 v1.20.0 [21] function filterAndTrim with a maximum number of expected errors (maxEE) equal to 1. Values ranging from 2 to 10 were tested for the truncQ parameter in filterAndTrim, which truncates reads at the first instance of a quality score ≤truncQ. The optimal value was determined to be the value that maximized the number of reads used for haplotype calling [10]; the haplotypes that were output when using this value of truncQ were used for all subsequent analyses. Next, the learnErrors function was used to learn error rates, the dada function was used to remove sequencing errors and identify haplotypes, and the removeBimeraDenovo function was used to remove chimeras. All haplotypes returned by DADA2 were included for analysis.

## Categorization of haplotypes

We define a haplotype as a unique sequence returned by DADA2 (as described above). For each locus in each sample, we further categorized each haplotype returned by DADA2 into one of three groups:

1. Expected haplotype: A haplotype with an identical sequence to that of a template sequence (reference haplotype) expected to be observed in the sequenced library. These were considered *true positive haplotypes*.

2. <u>Haplotype arising from systematic error</u>: A haplotype with a sequence or read depth that we did not expect to observe in the sequenced library, but which was observed across all three replicates for at least one density bin. These were suspected to be truly present owing to either inadvertent introduction to mixtures during gDNA preparation or the presence of multiple haplotypes in the original source gDNA. In most cases, the sequence of these haplotypes was that of a reference strain that was not expected to be present in the mixture, supporting the former over the latter hypothesis. Haplotypes arising from systematic error were removed from the analysis prior to screening for optimal thresholds for haplotype censoring, as we suspected that these template strains were truly present in the library that was sequenced and therefore shouldn't be expected to be corrected by applying filtering criteria.

3. <u>Haplotype arising from random error</u>: A haplotype that we did not expect to observe in the sequenced library, and that was not consistently present across replicates for any mixture-density combination. These were considered *false positive haplotypes*.

## Identification of optimal thresholds for haplotype censoring

We evaluated the efficacy of four common metrics used to censor haplotypes: i) the depth of reads within a sample supporting a haplotype (read depth), ii) the proportion of reads within a sample supporting a haplotype (read proportion), iii) the ratio of abundances of pairs of haplotypes within a sample with a Hamming distance of one (read ratio), and iv) the length difference of the returned haplotype relative to that of the expected reference strain (length difference). As mentioned above, haplotypes arising from systematic error were removed prior to evaluating these criteria. All reference strain haplotypes for all loci were identical in length to the 3D7 haplotype, except one *msp7* haplotype that was 3 base pairs shorter. Thus, we defined this censoring criterion as follows: the difference in length between the observed haplotype and the 3D7 reference haplotype must be equal to 0, -3, or 3 (i.e. one codon may be inserted or deleted). For the other 3 censoring criteria, we used Youden's J statistic to identify optimal thresholds across all possible thresholds of the criterion and corresponding confidence intervals with the coords and ci.coords functions from the R package pROC v1.18.0 [22]. Because the importance of retaining true positive haplotypes vs. removing false positive haplotypes varies depending on the use case, this statistic was computed using three different ways to weight false negative vs. false positive classifications: equal weight to false negatives and false positives, 2x the weight to false negatives, and 2x the weight to false positives. To evaluate censoring criteria, we used the optimal criteria based on false negatives having 2x the cost of false positives.

## Risk factor analysis for missing haplotypes

In order to identify what factors were associated with the failure to recover a haplotype from a mixture, we performed a bivariate and multivariate logistic regression of risk factors for haplotype missingness in R using the glmer function in lme4 v1.1.32 [23]. Missing haplotypes were defined as those that were not observed in the sample prior to the application of any haplotype censoring criteria. The outcome was the presence or absence of the haplotype in the un-censored haplotypes, and risk factors were target, starting proportion of the reference template strain, read depth (per 10,000 reads), parasite density, and expected number of distinct haplotypes present in the sample. A random intercept was included for each mixture-density combination. Low-density mix C samples were excluded from this analysis as they exhibited signatures of contamination from a high-density sample.

## Clinical sample analysis

Ten *P. falciparum*-positive DBS collected in a field study in Webuye, Kenya that were previously sequenced at the *ama1* and *csp* loci [2] were sequenced at the *msp7*, *sera2*, and *trap* loci. These samples were selected from those that were high-density and had MOIs >1 at both *ama1* and *csp* loci (using previously defined haplotype calls and censoring criteria [2]). Haplotypes for newly sequenced loci were called with the pipeline described above, using the same method as for *ama1* and *csp*. All haplotypes were censored using the identified optimal censoring criteria.

## Ethical statement

The field study in which the clinical samples were collected was approved by institutional review boards of Moi University (2017/36) and Duke University (Pro00082000). All participants or guardians provided written informed consent, and those over age 8 years provided additional assent. Enrollment began 1 July 2017, and as an open cohort, is ongoing.

## Data analysis and visualization

All data were analyzed and visualized using R v4.2.1 [24] in RStudio v2022.12.0+353 [25] with the following packages: msa v1.28.0 [26], tidyverse v2.0.0 [27], readxl v1.4.2 [28], ape v5.7.1 [29], regentrans v1.0.0 [30], reshape2 v1.4.4 [31], scales v1.2.1 [32], cowplot v1.1.1 [33], ggupset v0.3.0 [34], broom.mixed v0.2.9.4 [35], ggpmisc v0.5.2 [36], ggpubr v0.6.0 [37], and ggtext v0.1.2 [38]. From Pf6k [39] VCF files, we extracted and tallied the variant positions that passed filtering (i.e. FILTER = PASS) in the amplified portion of each marker gene. We compared read depths of true and false positive haplotypes, and median multiplicities of infection, using a Wilcoxon test, and number of haplotypes censored by density using a Fisher's exact test. Code and data to recreate the analyses and figures in this manuscript can be found at https://github.com/duke-malaria-collaboratory/haplotype_recovery_experiment.

## Results

### Mixtures, reference strains, deep sequencing, and haplotype calling

We sequenced five previously-developed AmpSeq marker genes: *ama1*, *csp*, *msp7*, *sera2*, and *trap* (Table 2), and generated for sequencing 6 mock infections harboring mixtures of gDNA from between 1 and 10 distinct parasite reference strains (Fig 1A) to approximate the polygenomic nature of many infections in high-transmission areas. Not all marker genes were unique to a strain; a total of 37 distinct haplotypes were present across the 10 strains and 5 markers.

**Table 2. Marker gene characteristics.**

| Target | Stage expressed | 3D7 gene ID | Chromosome | 3D7 coordinates amplified* | Sequence length | 3D7 GC content | Number of Pf6k variant positions** |
|--------|-----------------|-------------|------------|-----------------------------|-----------------|----------------|--------------------------------------|
| *ama1* | Blood | PF3D7_1133400 | 11 | 1294312–1294613 | 300 | 27% | 49 (16%) |
| *csp* | Liver | PF3D7_0304600 | 03 | 221351–221640 | 288 | 29% | 53 (18%) |
| *msp7* | Blood | PF3D7_1335100 | 13 | 1419236–1419567 | 330 | 25% | 53 (16%) |
| *sera2* | Blood | PF3D7_0207900 | 02 | 320762–321022 | 259 | 41% | 62 (24%) |
| *trap* | Liver | PF3D7_1335900 | 13 | 1465058–1465379 | 320 | 31% | 46 (14%) |

\* Coordinates correspond to those from PlasmoDB [40].

\*\* Includes all variants that passed filtering in the amplified region.

Pairwise single nucleotide variant (SNV) distance varied between strains and markers (median: 4, range: 0–15; Fig 1B).

For each marker gene, each of the 6 mixtures was sequenced from dilution pools corresponding to low (<1.5 genomes/μL), medium (1.5–75 genomes/μL) and high (≥75 genomes/μL) parasite density bins in triplicate, tallying to 1365 expected haplotype occurrences across 270 sequenced samples. We obtained analyzable reads for 257/270 samples, with differences in the absolute yield of read counts between low (4.3 million), medium (8.0 million), and high (10.2 million) density samples. This general observation held for each individual marker, save for *trap* and *msp7* which returned moderate read amounts irrespective of parasite density bin (Fig 1C). Overall, we observed across the five loci and 257 samples 1292 haplotype occurrences (Fig 1D), for which the median read depth was 1542. The haplotypes returned were identical when reads were mapped to either the 3D7 or to the HB3 reference genome.

## False positive haplotypes

We first investigated false positive haplotype occurrences across samples. Within each sample, we categorized each observed haplotype as expected to be present in the sample (true positive, n = 859/1292, 66%), likely cryptically present in the original mixture (systematic error; n = 254/1292, 20%), or likely arising from random error (false positive, n = 179/1292, 14%) (Fig 2A). Only 1% of reads that passed filtering supported haplotypes that were categorized as false positives. We observed this trend of proportionately few reads supporting proportionately more false positive haplotypes across both markers and parasite density bins (Fig 2B). Furthermore, the percentage of false positive haplotypes was relatively similar across parasite density bin (12–16%), although for *ama1* and *sera2*, there were fewer false positive haplotypes for low-density templates (Fig 2C). False positive haplotypes were often not the correct sequence length, were often only one nucleotide different from a reference sequence in the sample (Fig 2C), and had lower read depths than haplotypes we expected to observe (median = 104 vs. 2393, Wilcox p < 0.001; Fig 2D).

## Evaluating haplotype censoring criteria

We next evaluated, in our dataset, the effectiveness of four important threshold criteria typically applied to remove false positives from AmpSeq data: read depth, read proportion, read ratio of similar haplotypes, and haplotype length. The optimal thresholds had large confidence intervals and varied depending on how much weight was given to false positive vs. false negative haplotypes (Fig 3A–3C). Prioritizing the inclusion of true positive haplotypes over the removal of false positive haplotypes, optimal thresholds were 275 for read depth (95% CI: [204–420]; sensitivity = 0.95 [0.90–0.99]; specificity = 0.52 [0.46–0.68]), 0.007 for read proportion (95% CI: [0.005–0.014]; sensitivity = 0.97 [0.91–0.99]; specificity = 0.54 [0.47–0.69]), and 0.21 for read ratio (95% CI: [0.09–0.36]; sensitivity = 0.82 [0.72–0.93]; specificity = 0.67 [0.44–0.67]). Using these criteria, across all targets 975/1292 (75%) haplotype occurrences remained corresponding to 59/124 (48%) distinct haplotypes, yielding at least one uncensored haplotype in 254/257 (99%) samples. Specifically, these thresholds censored 148/179 (83%) random error haplotypes, 102/254 (40%) systematic error haplotypes, and 67/859 (8%) expected reference haplotypes (Fig 3D and 3E). Of the 179 random error haplotypes, 75% fell under the read threshold, 54% fell under the proportion threshold, 30% fell under the within-sample ratio threshold, and 28% had a length different than the reference strains. Furthermore, for all markers but *trap*, fewer false positive haplotypes were successfully censored in lower parasite density bins (Fisher's exact p < 0.01, Fig 3F), yielding more false positives post-censoring in low- (11) compared to medium- (6) and high-density (0) parasite bins. Of the censored true

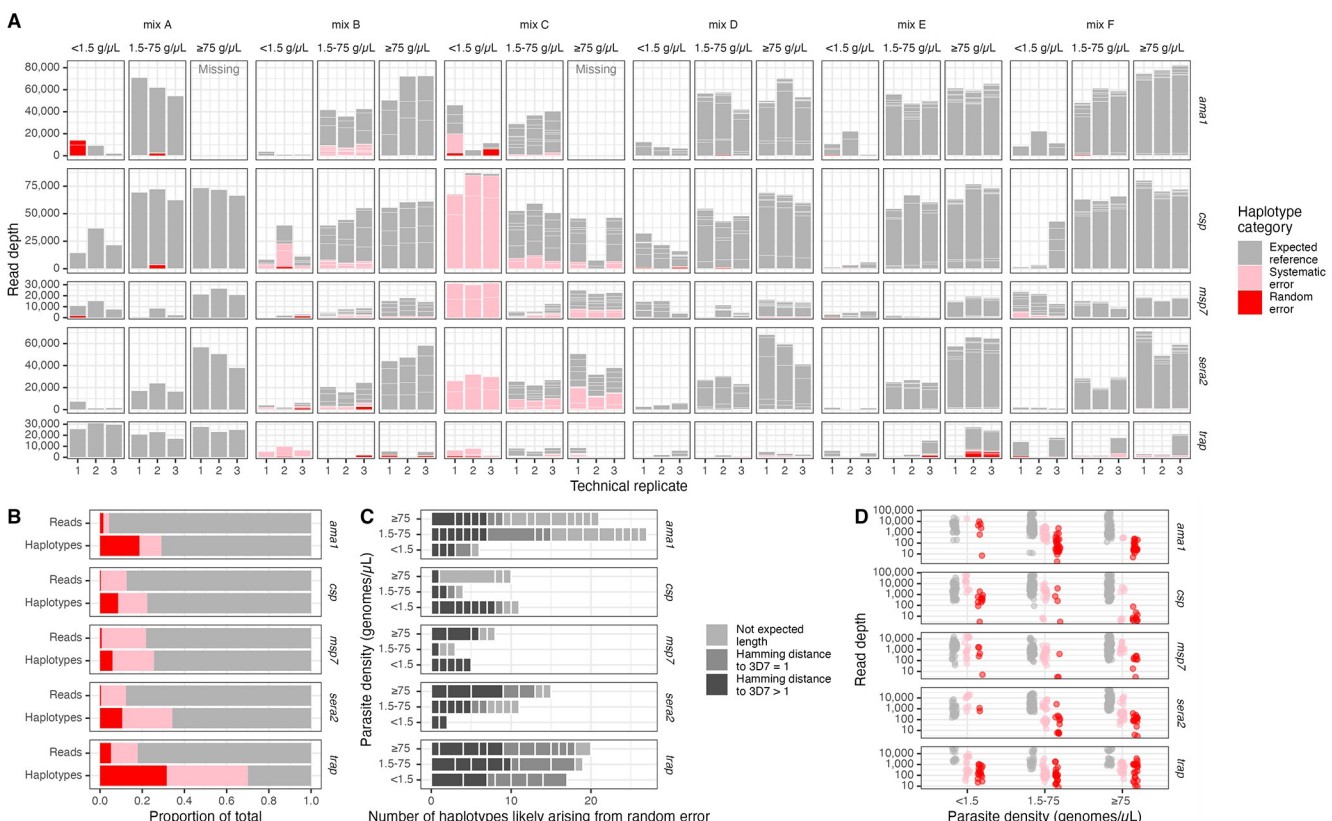

**Fig 2. Overview of false positive haplotypes. (A)** Sample-level haplotype overview. Stacked boxes in each column represent observed haplotypes from reads that passed filtering, categorized as those expected in the reference (gray), arising from systematic error (pink), or from random error (red). Box heights indicate the number of reads supporting the haplotype. **(B)** Proportion of reads and of haplotypes by marker categorized as expected reference, systematic error, and random error. **(C)** False positive haplotypes by marker categorized by unexpected length and by SNV distance to the 3D7 reference sequence. Hamming distances were only computed for haplotypes identical in length to the 3D7 reference sequence. **(D)** Read depth for expected (gray), systematic error (pink) and random error (red) haplotypes by parasite density bin and by marker. g/μL = genomes/μL.

positive haplotypes, over half (39/67; 58%) were from high-density templates, and only 5/67 (7%) made up ≥10% of the original mixture (Fig 3G).

## Inter-replicate variability

To evaluate the consistency with which haplotypes were returned, we measured inter-replicate variability post-censoring. Overall, 58% of haplotypes were observed in all 3 replicates, 18% in 2 replicates, and 24% in 1 replicate. Haplotypes were more consistently returned in all three replicates for high-density samples (76% of the time) compared to medium- (61% of the time) and low-density samples (30% of the time) (Fig 4A). Consistent with this, in high-density samples Jaccard distances between replicates were higher (median = 1, IQR = 0.2) compared to medium- (median = 0.83, IQR = 0.5) and low-density samples (median = 0.5, IQR = 0.75) (Fig 4B).

## Missing haplotypes

Of the 1365 haplotype occurrences expected to be present across all samples, we did not recover 477 (35%). Thus, we next investigated factors associated with missing haplotypes. As expected, haplotype proportion within a sample was inversely associated with missingness,

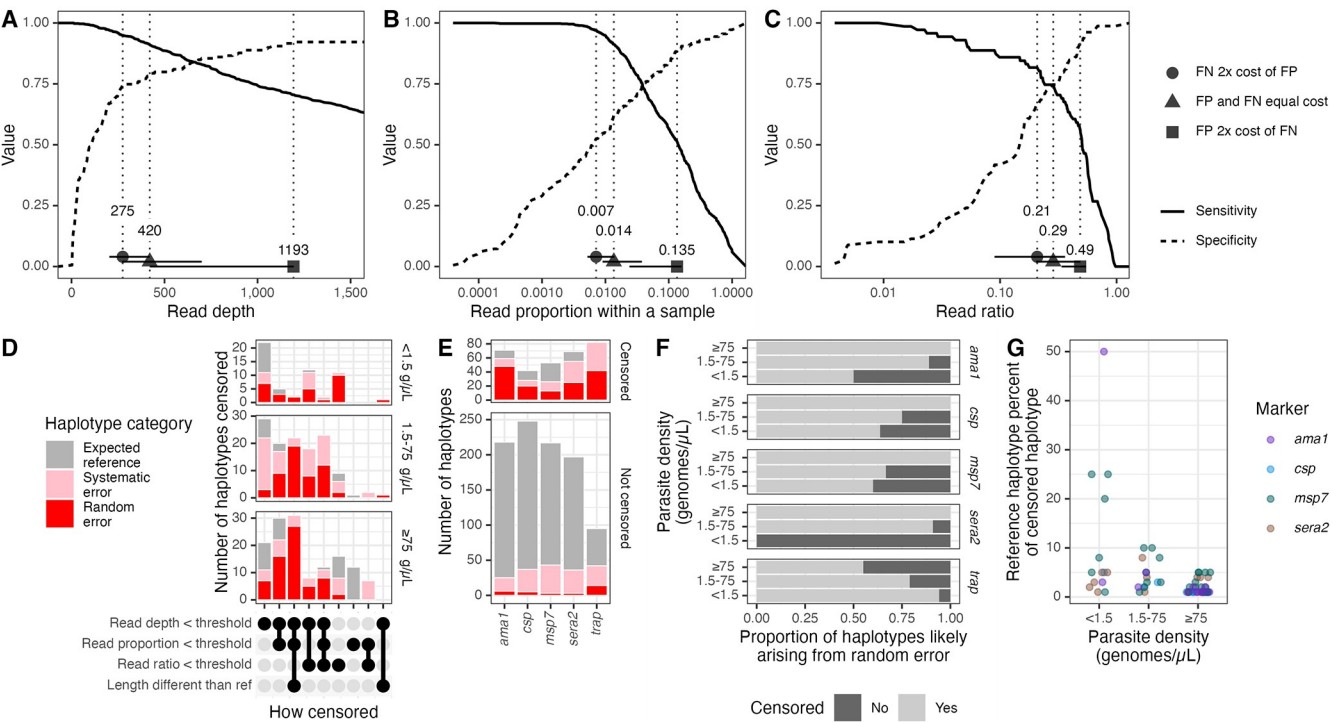

**Fig 3. Optimization and application of censoring criteria.** (A-C) Sensitivity and specificity across ranges of tested thresholds for haplotype (A) read depth, (B) read proportion, and (C) ratio within a sample between haplotypes with a Hamming distance of 1. (D) Count of censored haplotypes by the criterion by which they were censored and by density of parasites in DBS sample. The majority of censored haplotypes were non-reference haplotypes and fell under the identified read depth threshold. (E) Numbers of censored and uncensored haplotypes by haplotype category and by marker. (F) Proportion of uncensored (light grey) and censored (dark grey) haplotypes likely arising from random error, by parasite density bin and marker. (G) Reference haplotype percent of censored haplotypes. No reference haplotypes were censored out for *trap*. g/µL = genomes/µL. FN = false negative; FP = false positive. g/µL = genomes/µL.

with each increase of 0.01 in proportion associated with a 4% reduction in the likelihood of being missed (OR: 0.96, 95% CI: 0.96–0.97), even when controlling for marker, density bin, number of reads in the sample, and expected number of haplotypes (Fig 5A; Table 3). Additionally, for all markers except *trap*, <15% of haplotypes were missed from high-density

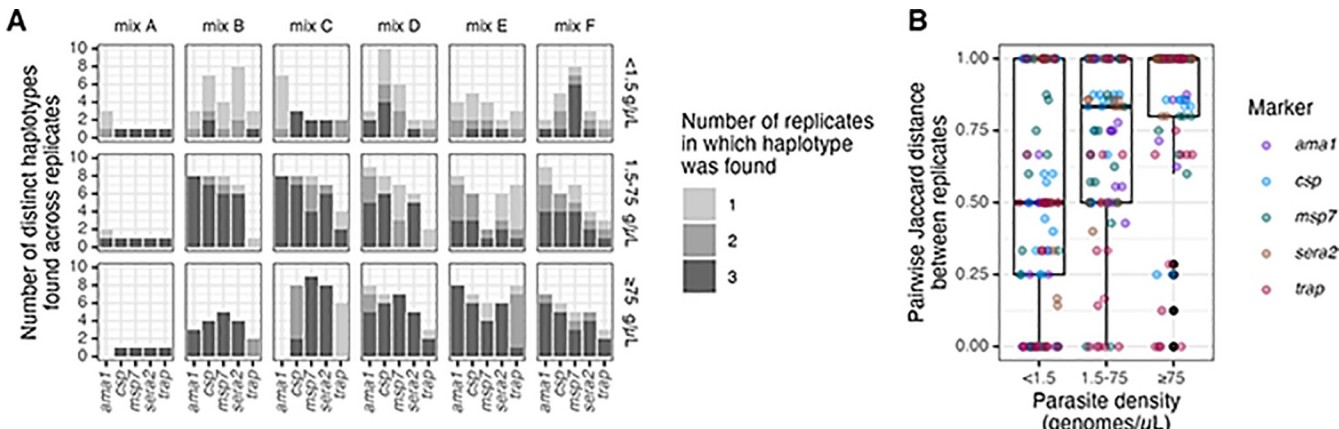

**Fig 4. Inter-replicate variability.** (A) Number of replicates in which each haplotype was found (color) by mix, parasite density bin, and target. (B) Pairwise Jaccard distance between replicates by parasite density bin, colored by marker. g/µL = genomes/µL.

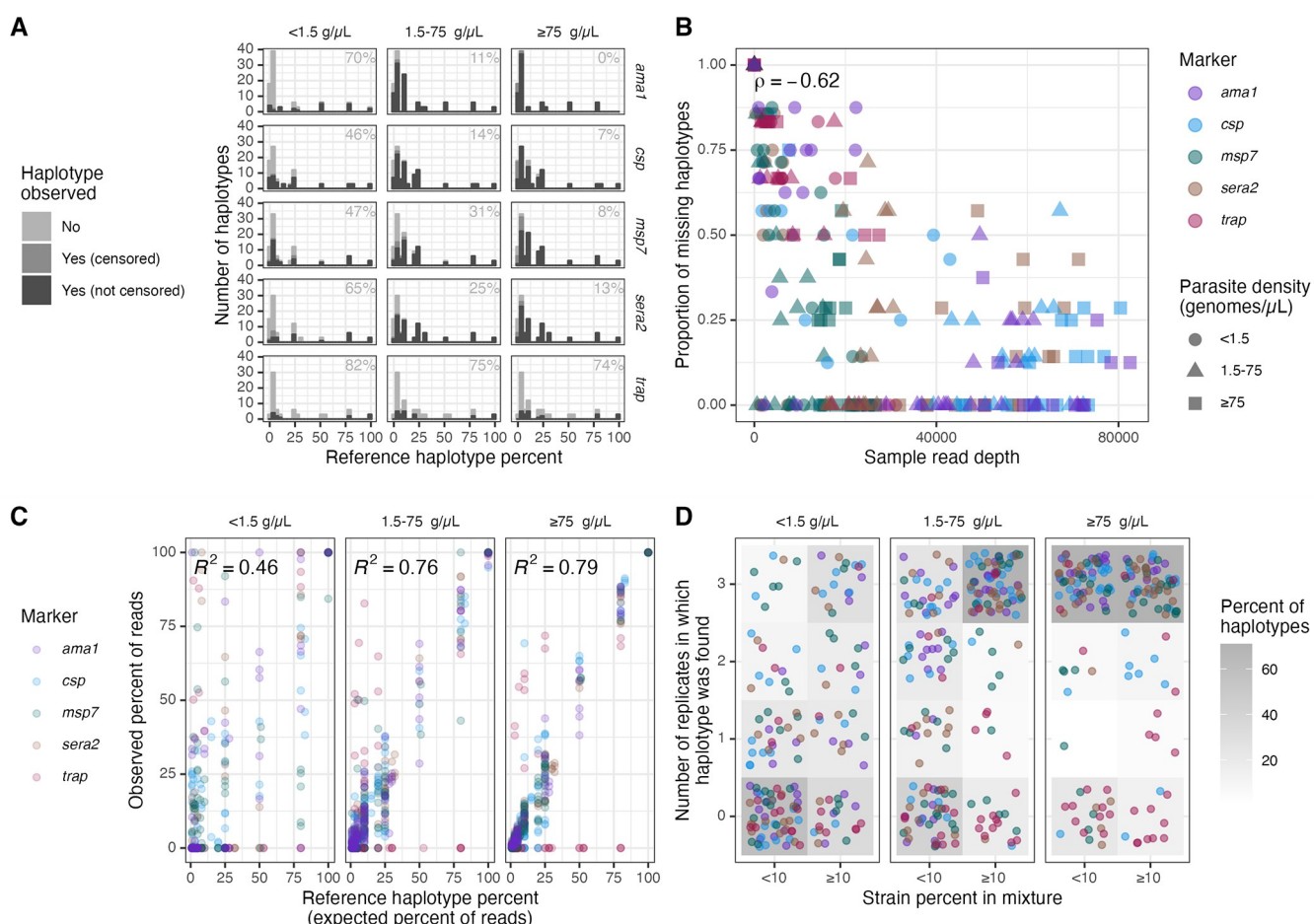

**Fig 5. Summary of missing haplotypes. (A)** Numbers of missing haplotypes (light grey), observed but censored haplotypes (medium grey), and observed haplotypes (dark grey) in individual samples by marker and parasite density bin. The number in each facet indicates the percentage of missing haplotypes. All subsequent panels in this figure consider observed but censored haplotypes as missing. **(B)** Correlation between the overall read depth of a sample and proportion of all expected haplotypes within a mixture that were not successfully recovered. Color indicates marker, and shape indicates parasite density. Spearman's rho = -0.62. **(C)** Correlation between proportions of expected and observed haplotypes within individual samples by parasite density bin, colored by marker. **(D)** Number of replicates in which the haplotype was found by binned strain percent in the original mixture (present at <10% or ≥10%). Each point is a haplotype colored by marker. The grey color beneath the points indicates the percent of haplotypes across all targets and mixtures in a given strain percent bin that were observed in the corresponding number of replicates. Low-density mix C samples were excluded from this figure as they exhibited signatures of contamination from a high-density sample. g/µL = genomes/µL.

samples, while >45% were missed from low-density samples (Fig 5A). Overall read depth for a sample was negatively correlated with the proportion of haplotypes that were missing from the sample (Spearman's rho = -0.62; Fig 5B). Furthermore, within a sample, observed and expected read proportions were correlated, although there was high stochasticity, particularly for the low-density samples (Fig 5C). Finally, in high-density samples only 30/166 (18%) haplotypes were not recovered in any replicates, while in low-density samples 78/158 (49%) were not recovered in any replicates (Fig 5D).

## Estimating multiplicity of infection based on marker haplotype diversity

We next compared the expected multiplicity of infection (MOI) to the observed MOI after censoring, with MOI expressed as the number of haplotypes observed at each individual

**Table 3. Risk factors for haplotype missingness.**

| Feature | Term | Bivariate | Multivariate |
|---|---|---|---|
| | | Odds Ratio (95% CI), p-value | Odds Ratio (95% CI), p-value* |
| Haplotype proportion (per 0.01 increase) | | 0.98 (0.97–0.98); p = 4.3e-11 | 0.96 (0.96–0.97); p = 6e-17 |
| Target | *ama1* | REF | REF |
| | *csp* | 0.76 (0.49–1.18); p = 0.23 | 0.90 (0.54–1.51); p = 0.7 |
| | *msp7* | 1.24 (0.81–1.89); p = 0.32 | 0.40 (0.23–0.68); p = 8e-04 |
| | *sera2* | 1.68 (1.1–2.57); p = 0.016 | 1.05 (0.63–1.77); p = 0.8 |
| | *trap* | 21.37 (13.02–35.08); p = 1e-33 | 6.13 (3.13–12.03); p = 1e-07 |
| Density, genomes/μL | ≥75 | REF | REF |
| | 1.5–75 | 1.62 (0.76–3.45); p = 0.21 | 1.47 (0.75–2.88); p = 0.3 |
| | <1.5 | 6.27 (2.87–13.69); p = 3.9e-06 | 3.88 (1.82–8.27); p = 5e-04 |
| Read depth (per 10,000 reads) | | 0.57 (0.53–0.62); p = 5.7e-40 | 0.61 (0.54–0.69); p = 3e-15 |
| Expected number of haplotypes | | 0.32 (0.24–0.44); p = 1.2e-12 | 1.08 (0.91–1.27); p = 0.4 |

\* Covariates included were haplotype proportion, target, parasite density, read depth, and expected number of haplotypes.

REF: reference group for each comparison. CI: Confidence Interval

marker. Relative to the expected MOIs, the observed MOIs were equal 29% (74/254) of the time, lower 61% (154/254) of the time, and higher only 10% (26/254) of the time. MOIs were more likely to be underestimated in low-density samples (median observed-expected MOI = -4 for low-density samples vs. -1 for medium- and high- samples, Wilcox p < 0.001; Fig 6A). We performed a similar comparison using 10 high-density *P. falciparum* infections collected as DBS through a recent field study in Western Kenya, in order to capture a broader naturally-occurring diversity of marker haplotypes [2]. Using the optimal censoring criteria defined

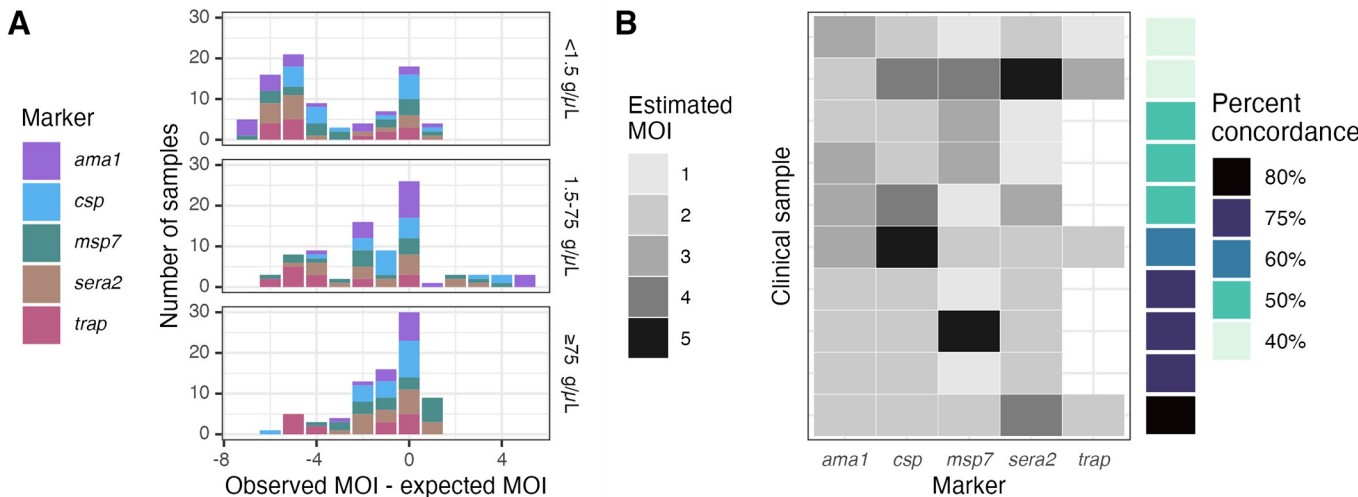

**Fig 6. Estimated multiplicities of infection (MOIs) based on each marker haplotype. (A)** Observed minus expected MOIs for mixtures post-censoring. **(B)** Estimated MOIs in clinical samples. Haplotypes were censored according to the optimal criteria identified above, giving false negatives 2x the cost of false positives. Not all *trap* clinical samples returned sequences. Percent concordance was computed for each sample as the percentage of markers for which the estimated MOI was equal to the mode MOI.

above, we observed 142 haplotypes across all samples and markers, of which 36 (25%) were censored. The range of MOIs was 1–5 for each marker. No marker consistently estimated the highest or lowest MOI, and the percent concordance ranged from 40% to 80% (Fig 6B).

## Discussion

AmpSeq is an increasingly popular tool for molecular epidemiologic studies of various pathogens including *P. falciparum* collected on DBS, which necessitates rigorous haplotype recovery from field samples. We prepared DBS containing mixtures of gDNA from reference *P. falciparum* strains, amplified and sequenced polymorphic segments of 5 common marker genes in triplicate, and quantified the performance of haplotype recovery using a range of metrics. We observed that high sample read depth was associated with enhanced recovery of most haplotypes present in the original sample, and that censoring criteria based on read depth, read proportion, read ratio, and haplotype length can effectively remove most false positive haplotypes while retaining most true positive haplotypes. Thus, for use-cases which involve high-density samples or samples sequenced at high read depth, rigorous recovery can be achieved for multiple markers.

Consistent with prior studies [2, 10, 12], we observed that the likelihood of haplotype recovery is enhanced by higher parasite density and by a larger proportion of an individual haplotype within a mixture. In particular, the consistency with which we observed haplotypes across replicates was higher in high-density samples compared to low-density samples. However, we further observed that, independent of parasite density and reference haplotype proportion, successful haplotype recovery was further associated with a higher overall sample read depth. The ability to recover haplotypes constituting a minority population within a parasitemia with an overall low density is an important goal for many use cases of AmpSeq. Namely, therapeutic efficacy studies of antimalarials use active case detection to screen for recurrence of parasites, and frequently capture low-density infections with multiple strains which must then be compared to those in the initial infection in order to distinguish reinfection from recrudescent infection [7]. Additionally, studies of transmission networks in highly endemic settings in which low-density, asymptomatic infections predominate also benefit from comprehensive profiling of strains within mixtures in order to ascertain parasite relatedness between hosts [2]. In these and similar use cases, the likelihood of detecting minority haplotypes can be improved by maximizing per-sample read depth, such as by limiting multiplexing and selecting maximal sequencing platform output.

We observed very different optimal censoring thresholds depending on how we weighted the relative importance of false positive and false negative haplotypes, which highlights the need to select censoring criteria suitable for the primary study objective. Penalizing false negative haplotypes more than false positive haplotypes yielded haplotype censoring criteria that still managed to remove most false positive haplotypes while retaining high sensitivity. Furthermore, these criteria were consistent with thresholds that others have used and reported in the literature (read depth: 204–420, read proportion: 0.005–0.014, read ratio: 0.09–0.36) [2, 12].

We observed inconsistency in performance between markers with respect to false positives, censoring, missingness, and MOI. Pre-censoring, false-positive haplotypes were rarely recovered for *msp7* but common for *ama1* and *trap*. However, post-censoring the number of false positives was relatively low for all markers but *trap*. Fewer haplotypes were recovered for *sera2* and *trap* overall. Furthermore, there was no consistent trend across a limited set of clinical samples of marker-specific MOI, suggesting that MOI estimates based upon a single marker may frequently underestimate the true MOI of a sample, as previously described [12]. Since most markers returned largely correct haplotype calls across a range of mixtures

and parasite density bins, choice of marker may depend not only on marker performance but also other factors such as the biological question of interest (e.g. transmission, vaccine development, etc.).

Despite controlled laboratory conditions, we observed signatures of both systematic and random error. Systematic error may have resulted from two different sources. First, it is possible that multiple haplotypes were present in the original template strains and were missed during Sanger sequencing, a limitation of this sequencing method. Second, systematic error could arise from contamination during gDNA extraction. Owing to the high-throughput manner of DBS processing, using 96-well plates, it is unfortunate but expected that we observe contamination in a small minority of samples included in a sequencing run. This highlights the importance of meticulous laboratory work and thoughtful controls, particularly because these haplotypes are less likely to be removed by censoring criteria owing to their presence in the original template. In contrast, random error may arise due to PCR stochasticity and polymerase error in low-input next-generation sequencing libraries [41]. This is also inevitable, and the censoring criteria described here successfully removed many haplotypes arising from these technical errors.

Our study had several limitations. First, we created the mixtures from gDNA rather than from intracellular DNA; therefore, the composition of the solution from which DNA was amplified was slightly less complex than that from clinical samples. However, as we extracted DNA from DBS, our results provide a closer approximation to clinical samples than previous studies. Second, we did not attempt to censor haplotypes arising from systematic error because the commonly used censoring criteria assessed here assume that false positive haplotypes arise from random rather than systematic error. Third, this study focused on in silico recovery of haplotypes, and replicates were drawn from the same gDNA extract pools. Thus, variability occurring due to extraction is not accounted for in these data. However, our results provide useful insight into variation and random errors occurring at the amplification and sequencing steps.

## Conclusions

We observed that *P. falciparum* haplotypes from multiple different targets can be successfully recovered from DBS, that in the majority of cases these haplotypes are recovered across replicates, and that censoring criteria already used by the community remove most false positive haplotypes while retaining high sensitivity. These observations can be used to guide analysis and interpretation both of *P falciparum* haplotypes recovered from DBS but also of other pathogens who share with malaria parasites high genetic variability, multiplicity of strains, and informative genetic markers.

## Supporting information

**S1 Checklist. Inclusivity in global research.**
(DOCX)

## Acknowledgments

We thank Jenna DeCurzio for helping with the preparation of samples for sequencing, as well as Laura-Leigh Rowlette and Fangfei Ye at the Duke University Sequencing & Genomic Technologies Shared Resource for performing sequencing and preliminary processing of sequenced reads. *P. falciparum* strains 3D7 (MRA-102, contributed by Daniel J. Carucci), FUP UGANDA-PALO ALTO (MRA-915, contributed by T. Sam-Yellowe), Dd2 (MRA-150G,

contributed by David Walliker), 7G8 (MRA-152G, contributed by David Walliker), HB3 (MRA-155G, contributed by Tom Wellems), K1 (MRA-159G, contributed by Dennis Kyle), V1/S (MRA-176G, contributed by Dennis Kyle), Tanzania (MRA-1169G, contributed by Michal Fried), FCB (MRA-309G, contributed by Tom Wellems), and FCR3/Gambia (MRA-731G, contributed by William Trager) were obtained from BEI Resources, NIAID, NIH.

## Author Contributions

**Conceptualization:** Zena Lapp, Christine F. Markwalter, Wendy Prudhomme-O'Meara, Steve M. Taylor.

**Formal analysis:** Zena Lapp.

**Funding acquisition:** Andrew A. Obala, Wendy Prudhomme-O'Meara, Steve M. Taylor.

**Investigation:** Zena Lapp, Elizabeth Freedman.

**Methodology:** Zena Lapp, Christine F. Markwalter.

**Project administration:** Andrew A. Obala, Wendy Prudhomme-O'Meara, Steve M. Taylor.

**Resources:** Andrew A. Obala.

**Software:** Zena Lapp, Kathie Huang.

**Supervision:** Andrew A. Obala, Wendy Prudhomme-O'Meara, Steve M. Taylor.

**Visualization:** Zena Lapp.

**Writing – original draft:** Zena Lapp, Steve M. Taylor.

**Writing – review & editing:** Zena Lapp, Christine F. Markwalter, Andrew A. Obala, Wendy Prudhomme-O'Meara, Steve M. Taylor.

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
