## [Decision Letter · Decision Letter 0]

26 Sep 2023

PGPH-D-23-01564

Analytic optimization of Plasmodium falciparum marker gene haplotype recovery from amplicon deep sequencing of complex mixtures

Dear Dr. Taylor,

Thank you for submitting your manuscript to PLOS Global Public Health. After careful consideration, we feel that it has merit but does not fully meet PLOS Global Public Health’s publication criteria as it currently stands. There are numerous serious concerns ranging from narrow focus to true definition of haplotype, high false positive rates, and many more. If you are willing to address these, we invite you to submit a revised version of the manuscript that addresses the points raised during the review process. Please note, your revised manuscript will likely undergo another round of review prior to making a final decision.

We look forward to receiving your revised manuscript.

Kind regards,

Nirbhay Kumar, PhD

Academic Editor

Journal Requirements:

2. We ask that a manuscript source file is provided at Revision. Please upload your manuscript file as a .doc, .docx, .rtf or .tex.

Additional Editor Comments (if provided):

Reviewers' comments:

Reviewer's Responses to Questions

**Comments to the Author**

1. Does this manuscript meet PLOS Global Public Health’s publication criteria? Is the manuscript technically sound, and do the data support the conclusions? The manuscript must describe methodologically and ethically rigorous research with conclusions that are appropriately drawn based on the data presented.

Reviewer #1: Partly

Reviewer #2: Partly

2. Has the statistical analysis been performed appropriately and rigorously?

Reviewer #1: No

Reviewer #2: I don't know

3. Have the authors made all data underlying the findings in their manuscript fully available (please refer to the Data Availability Statement at the start of the manuscript PDF file)?

Reviewer #1: No

Reviewer #2: No

4. Is the manuscript presented in an intelligible fashion and written in standard English?

Reviewer #1: Yes

Reviewer #2: Yes

5. Review Comments to the Author

Reviewer #1: Review for PLoS Public Health

MS# PGPH-D-23-01564

Title: Analytic optimization of Plasmodium falciparum marker gene haplotype recovery from

amplicon deep sequencing of complex mixtures

Authors: Lapp et al.

Summary:

This is a focused paper attempting to characterize the ability of the AmpSeq approach using next-generation sequencing to accurately characterize and quantify haplotype diversity in Plasmodium falciparum, a vector for malaria. The study adds to an extensive body of such studies by increasing the ‘community’ of diversity of mock community to as many as 10 (but the stated diversity is up to 15) with previous studies up to 3. Additionally, the study focuses on the utility of getting such data from blood spots. Thus, there is an advance, but it is in a very narrow context. The authors do not attempt to place their research in the broader context of haplotype calling abilities. The general conclusion seems to be that approaches are good even though they seem to miss 35% of the haplotypes. The false positive and true positive rates are problematic because the authors think they have true positives in the next-generation data compared to Sanger sequence data, but then eliminate these results from downstream analyses. They fail to acknowledge that they simply do not know the ‘truth’ in these data. Thus, the calculations of ‘true’ and ‘false’ positives is problematic. The paper would be enhanced with some simulation data where ‘truth’ is known. Alternatively, the authors might consider submitting to a more malaria focused journal. As it is, the haplotype study is not placed in a significantly broad enough context to be of general interest to the broader readership of this journal. Below, I have a number of comments I hope the authors will find useful in revising their work.

The conclusion in the abstract, that haplotypes can be successfully recovered, seems at odds with the data presented showing a failure to recover 35% of the haplotypes in this controlled environment.

The conceptual framework of this paper seems very narrow, calling haplotypes in this one species. I understand this focus and its public health relevance, but the paper misses an opportunity to relate this work to the broader concept of haplotype calling that is also key in other infectious diseases, e.g., HIV, HCV, COVID-19, etc. Furthermore, there are extensive simulation studies that offer insights into the generally poor performance of haplotype callers (e.g., https://doi.org/10.1016/j.meegid.2020.104277). The paper would be better served and more broadly applicable if it related the study findings to this broader context and more extensive literature on haplotype callers. If the authors prefer to keep the focus on this one species, that’s fine, but then I think this is not the right venue for this paper. For the broader readership of PLoS Public Health, it is important to provide linkages and insights beyond this one species. This is especially important this this sort of study has already been done in this species. The authors extend the insights by having a larger mock community and getting data from blood spots, but in the end, the conclusions are very similar.

Line 117, please provide BioProject number.

Line 121, GitHub link does not work, thus, the pipeline is not available for review.

Line 128, the authors use P.f. 3D7 as the reference genome to which why map reads and call haplotypes. Other studies (not cited) have shown that the reference genome can make a big difference in the number of haplotypes called. The authors might explore this aspect. There are a large number of P.f. genomes available. The choice of 3D7 is not justified. Assuming the authors chose this genome for a good reason (it is geographically and/or temporally relevant to ongoing malaria outbreaks (?), they might do something like estimate a phylogeny of the P.f. genomes available on GenBank (~34) and then choose another that is evolutionarily distant from the 3D7 and rerun their pipeline to see the impact on haplotype calls. This should be a pretty straightforward analysis that is just a change in the reference genome and then re-running the established pipeline, doesn’t cost extra lab supplies, and would provide some additional insights with respect to dependence of inferences on the reference genome choice for haplotype calling.

Line 145, the systematic error definition seems to be measuring the error rate in the Sanger sequencing approach to estimating ‘truth’ in haplotypes. This is the fundamental flaw with these sorts of empirical studies compared to computer simulation, you don’t actually know the ‘truth’ and therefore true/false positive rates are really hard to get to. Rather than ignoring these data, I recommend the authors report these results as well as some direct comparison of Sanger versus DBS for identifying haplotypes. This makes the whole haplotype censoring (lines 158-177) statistics strange because you clearly don’t have reasonable estimates of ‘true positives’ if you have a bunch of true positives that you are throwing out of the analysis because you didn’t see them with Sanger sequencing. This whole like of logic seems flawed to me.

Lines 179-188 – what is the ‘risk factor’ for Sanger sequencing relative to the haplotypes that looked real with the DBS?

Lines 216 – 219 are all repeat of the methods.

Lines 224 – 226 are method repeats as well.

Reviewer #2: Understanding how next generation sequencing data should be filtered to minimize false positives is important for the analysis of malaria parasite field samples. Here the authors have conducted a careful mock study in which they mixed proportions of parasites with known variants together at different proportions to determine if they could recover the expected mixtures and haplotypes from their sequencing data. The authors present an extensive statistical analysis and show that extensive censoring of the data improves the results.

Overall, this is most likely a useful and robust study and the authors conclusions do seem valid. However, as presented, I have doubts about its overall utility to the clinicians and public health researchers who will be the users and readers of this article. A major problem is related to poor standard scientific communication and failure to adhere to scientific publishing standards that involve providing the location of supporting data needed to check the authors’ conclusions and as well as descriptions of the accompanying datasets.

I have indicated below a list of places where the scientific communication could be improved but it is not a comprehensive list by any means. I would recommend that the authors make a good effort to make sure that all of their supporting data is well labeled with callouts and extensive descriptions before this is resubmitted for consideration.

Major concerns

The supplemental data is poorly annotated. The authors direct the reader to a website which contains some of the data, but the various files are not referenced at all within the text and it difficult to try to sort out what they did or what the files mean. The lack of callouts to supplemental tables is nonstandard and makes review difficult. For example, what exactly is dada2_long.csv? I can sort of figure it out, but it is not really the reviewers’ responsibility to spend hours deconvoluting the authors work and to try to interpret what is contained in poorly labeled tables. It really is responsibility of the corresponding author to make sure this is done. The various intermediate files on the website should be given table numbers and there should be explicit callouts to these table numbers throughout the text. For example, the authors write “The products of each were Sanger sequenced to determine the reference sequence for each strain and the results of the sequencing are given in Table SY.” These data showed that our results were similar (or identical?) to sequences deposited in PlasmoDB for X of the strains, correctly calling each of the X expected variants. In another example, the authors write in the legend to Figure 1, “Pairwise single nucleotide variant (SNV) distances between reference haplotypes of each of the marker genes obtained by Sanger sequencing” but then fail to mention that this raw data appears to be contained in a file called “ref_pairwise_dists.csv” on their github site. Pairwise distance ref is also a confusing name. What they actually show is the number of SNVs distinguishing two clones/strains across all queried markers.

The authors should not duplicate material that is in the main text either. I see a file called “target overview” which seems to be an exact clone of Table 2. It is work for the reviewer to go through the tables and putting duplicated tables into the supplement wastes the reviewers’ time. The figures are also duplicated on the website.

Overall. I am a bit confused about how the authors define a haplotype. For a given strain such as FCR3, given that this is a haploid organism, there should be only 1 haplotype, right? So if you properly sequence 10 haploid clonal strains, you should get exactly 10 reference haplotypes, even if you sequence five different marker genes for each. The authors mention that they have 34 reference haplotypes. Does this mean there were new alleles that were not detected by Sanger sequencing? Or are they creating extra artificial haplotypes by linking FCR3 ama1 to Dd2 csp? It would be easier to understand if the authors mentioned that there were X variable sites in the dataset with a total of Y different queried SNVs for the 10 strains. Overall, the focus on haplotypes instead of called variants makes it this more difficult to understand. I assume 34 is a subset of the possible 50, reduced by the fact that some strains (FCR3 and FCB) appear to be identical to one another in th

---

## [Decision Letter · Decision Letter 1]

5 Mar 2024

PGPH-D-23-01564R1

Analytic optimization of Plasmodium falciparum marker gene haplotype recovery from amplicon deep sequencing of complex mixtures

Dear Dr. Taylor,

Thank you for submitting your manuscript to PLOS Global Public Health. After careful consideration, we feel that it has merit, however, we invite you to submit a revised version of the manuscript that addresses the points raised during the review process.

We look forward to receiving your revised manuscript.

Kind regards,

Nirbhay Kumar, PhD

Academic Editor

Journal Requirements:

Additional Editor Comments (if provided):

Reviewers' comments:

Reviewer's Responses to Questions

**Comments to the Author**

1. If the authors have adequately addressed your comments raised in a previous round of review and you feel that this manuscript is now acceptable for publication, you may indicate that here to bypass the “Comments to the Author” section, enter your conflict of interest statement in the “Confidential to Editor” section, and submit your "Accept" recommendation.

Reviewer #1: All comments have been addressed

2. Does this manuscript meet PLOS Global Public Health’s publication criteria? Is the manuscript technically sound, and do the data support the conclusions? The manuscript must describe methodologically and ethically rigorous research with conclusions that are appropriately drawn based on the data presented.

Reviewer #1: Yes

3. Has the statistical analysis been performed appropriately and rigorously?

Reviewer #1: Yes

4. Have the authors made all data underlying the findings in their manuscript fully available (please refer to the Data Availability Statement at the start of the manuscript PDF file)?

Reviewer #1: No

5. Is the manuscript presented in an intelligible fashion and written in standard English?

Reviewer #1: Yes

6. Review Comments to the Author

Reviewer #1: The authors have done a fine job responding to my suggestions from the first review. The GitHub repo is now accessible. However, the BioProject is not. That will need to be fixed before publication. Otherwise, they've done a nice job of tightening up the manuscript.

7. PLOS authors have the option to publish the peer review history of their article (what does this mean?). If published, this will include your full peer review and any attached files.

**Do you want your identity to be public for this peer review?** For information about this choice, including consent withdrawal, please see our Privacy Policy.

Reviewer #1: No

---

## [Editor Report · Decision Letter 2]

30 Apr 2024

Analytic optimization of Plasmodium falciparum marker gene haplotype recovery from amplicon deep sequencing of complex mixtures

PGPH-D-23-01564R2

Dear Dr Taylor,

We are pleased to inform you that your manuscript 'Analytic optimization of Plasmodium falciparum marker gene haplotype recovery from amplicon deep sequencing of complex mixtures' has been provisionally accepted for publication in PLOS Global Public Health.

Best regards,

Nirbhay Kumar

Academic Editor